# Quill Mites of the Subfamily Syringophilinae (Acariformes: Syringophilidae) Parasitising Starlings (Passeriformes: Sturnidae) [note 1]

**DOI:** 10.3390/ani14152239

**Published:** 2024-08-01

**Authors:** Iva Marcisova, Maciej Skoracki, Milena Patan, Martin Hromada, Bozena Sikora

**Affiliations:** 1Laboratory and Museum of Evolutionary Ecology, Department of Ecology, Faculty of Humanities and Natural Sciences, University of Prešov, 08001 Prešov, Slovakia; marcisova.iva@gmail.com (I.M.); hromada.martin@gmail.com (M.H.); 2Department of Animal Morphology, Faculty of Biology, Adam Mickiewicz University, Uniwersytetu Poznańskiego 6, 61-614 Poznań, Poland; milpat@st.amu.edu.pl; 3Faculty of Biological Sciences, University of Zielona Góra, Prof. Z. Szafrana 1, 65-516 Zielona Góra, Poland

**Keywords:** Acari, birds, ectoparasites, new species, Sturnidae, Syringophilidae, taxonomy

## Abstract

**Simple Summary:**

This study provides detailed information on six previously described species of syringophilines associated with starlings and records new host and locality data. Additionally, three new species are described: *Aulonastus indonesianus* sp. n., *Aulonastus anais* sp. n., and *Syringophiloidus poeopterus* sp. n., from various starling hosts across Indonesia, Papua New Guinea, and Tanzania. The study reveals that species from the genera *Syringophiloidus* and *Syringophilopsis* exhibit a broad host range among passerine birds, suggesting a long-established symbiotic relationship before the global dispersal of starlings. In Europe, the Eurasian Starling hosts *Krantziaulonastus*, while in Africa, related niches are occupied by Picobiinae mites. The research highlights the intricate coevolutionary dynamics between quill mites and their avian hosts.

**Abstract:**

Quill mites of the family Syringophilidae are widely distributed parasites of birds inhabiting the interior of feather quills. In this paper, we provide detailed information on the host spectrum and distribution for six previously described species of syringophilines associated with starlings with new host and locality records. Additionally, we describe three new species: *Aulonastus indonesianus* Marcisova, Skoracki, and Patan sp. n. from the Common Hill Myna *Gracula religiosa* Linnaeus in Indonesia (Java) and the White-necked Myna *Streptocitta albicollis* (Vieillot) in Indonesia (Celebes); *Aulonastus anais* Skoracki and Patan sp. n. from the Golden Myna *Mino anais* (Lesson) in Papua New Guinea; and *Syringophiloidus poeopterus* Skoracki and Patan sp. n. from the Abbott’s Starling *Poeoptera femoralis* (Richmond) in Tanzania. Finally, we explore the host–parasite interactions within the system comprising starlings and syringophiline mites.

## 1. Introduction

Mites of the family Syringophilidae Lavoipierre, 1953, belong to widely distributed parasites of birds, spending their entire lives inside feather quills. In this unique habitat, they feed, undergo their entire development, and copulate [1,2,3,4,5]. The dispersing forms are adult fertilised females, which move to a new host during the breeding season, from parents to offspring (vertical transfer). Horizontal transfer of parasites can also occur during host mating or through frequent contact between host individuals, especially among social birds or between predators and their prey [6,7,8,9]. The infestation rate of the host population by Syringophilidae mites is undoubtedly dependent on the host’s behaviour; it is relatively high in social species (reaching up even to 70%) and low in solitary species [10,11,12,13]. It is worth noting that in the field studies provided in Poland, the examined population of the Eurasian Starling was infested by quill mites on a quite high level (the prevalence = 54%) [14].

Syringophilids, living and feeding on the tissue fluids of birds, are not considered to significantly reduce bird fitness [15]. However, our understanding of the harm caused by syringophilids is still incomplete. Veterinarians have observed clinical symptoms of feather picking in domestic birds attributed to quill mites, as reported in various studies [16,17,18]. Gritschenko [19] suggested that *S. bipectinatus* feeding induces itching, leading chickens to peck at affected feathers. This behaviour was thought to cause feather loss due to muscle tone relaxation, allowing new feathers to displace old, infested ones. Despite these findings, recent research shows no signs of quill mites causing skin or feather morphological changes in wild birds, even under heavy infestation [13,14,15]. On the other hand, the studies provided by Skoracki et al. [20] suggest that quill mites could act as vectors for the bacterium *Anaplasma phagocytophilum*, an obligate intracellular pathogen. These studies specifically found this pathogen in syringophiline mites, parasitising Eurasian Starling (Sturnidae) and Blackbirds (Turdidae). From an epidemiological perspective, the vertical transmission of syringophilids could potentially accelerate the spread of various avian diseases within bird populations. Since then, several other symbionts have been identified in quill mites, including multiple genetically distinct lineages of *Wolbachia* [21] and *Spiroplasma* [22].

The family Syringophilidae is divided into two subfamilies, Syringophilinae and Picobiinae, significantly differing from each other in terms of morphology, biology, and ecology [2,4,23]. Mites from the Syringophilinae subfamily, the subject of this article, exhibit high specificity towards their host groups. Each mite species typically parasitises a narrow host range, usually confined to hosts within a single order. For example, *Aulobia* and *Syringophilopsis* are found exclusively in birds of the order Passeriformes, while *Syringophilus* and *Colinophilus* parasitise Galliformes, and *Creagonycha* and *Niglarobia* are associated with Charadriiformes [4,5]. Additionally, Syringophilinae species are specific to the type of feather they inhabit. For instance, *Syringophiloidus* prefers secondary feathers, *Syringophilopsis* is found in primary and secondary feathers, *Neoaulonastus* inhabits secondary feathers and coverts, and *Aulonastus* occupies coverts and contour feathers [2,4,5].

The Syringophilinae subfamily currently includes 328 species grouped in 50 genera [24,25]. They have been found on host representatives belonging to many orders of neognathous birds (Neognathae), whereas in paleognathous birds (Paleognathae), they are known only among representatives of the Tinamiformes order. However, Skoracki et al. [26] suggest that their presence as representatives of this order is the result of a switch from hosts belonging to neognathous birds. The Passeriformes order has its own syringophiline fauna grouped into 12 genera [2,4,5]. Among this diversity, four genera of syringophilines have been recorded from starlings so far. In this paper, we provide detailed information on the host spectrum and distribution for all described species of syringophilines with new host and locality records. Additionally, we describe three new species within the genera *Aulonastus* and *Syringophiloidus* collected from starlings captured in Indonesia, Papua New Guinea, and Tanzania.

## 2. Materials and Methods

The mite material used in the present study was collected from dry bird skins housed in the ornithological collection of the Bavarian State Collection of Zoology, Munich, Germany (Staatliche Naturwissenschaftliche Sammlungen Bayerns SNSB-ZSM). Overall, we examined 44 species belonging to the family Sturnidae, of which only nine were infested by syringophiline mites, i.e., *Ampeliceps coronatus* Blyth, *Gracula religiosa* Linnaeus, *Mino anais* (Lesson), *Notopholia corrusca* (Normann), *Onychognathus morio* (Linnaeus), *O. nabouroup* (Daudin), *Poeoptera femoralis* (Richmond), *Sarcops calvus* (Linnaeus), and *Streptocitta albicollis* (Vieillot). From each infested individual, we collected one wing covert and 4–5 upper and under tail coverts. Mites were removed from wing covert and under-tail coverts using sharp tweezers. Before mounting, specimens were softened and cleared in Nesbitt’s solution at room temperature for three to four days, according to the protocol introduced by Krantz and Walter [27] and Skoracki [4].

Identification of mite specimens and drawing preparations were carried out with a light microscope (ZEISS Axioscope2™, Oberkochen, Germany) with differential interference contrast (DIC) optics and a camera lucida. All measurements in the descriptions are given in micrometres. The nomenclature for the idiosomal setation follows Grandjean [28], as adapted for Prostigmata by Kethley [29], leg setation is that of Grandjean [30], and general morphological terms follow Skoracki [4].

Specimen depositories are cited using the following abbreviations: AMU—Adam Mickiewicz University, Department of Animal Morphology, Poznań, Poland; SNSB-ZSM—Bavarian State Collection of Zoology, Munich, Germany.

The common and scientific names of birds are after Clements et al. [31] and del Hoyo et al. [32]. Zoogeographical regions are after Holt et al. [33] and Ficetola et al. [34].

## 3. Results

### 3.1. Descriptions

#### 3.1.1. *Aulonastus indonesianus* Marcisova, Skoracki and Patan sp. n.

Female, holotype (Figure 1 and Figure 2A–C). The total body length is 530 (555–595 in 45 paratypes). Gnathosoma. The infracapitulum is apunctate. Movable cheliceral digits are 130 (130–135) long. The stylophore is 175 (170–185) long, and the exposed portion of the stylophore is apunctate and 130 (130–140) long. Each medial branch of the peritremes has two chambers, and each lateral branch has seven chambers (Figure 2A). Idiosoma. The propodonotal shield is well sclerotised, apunctate, bearing bases of setae *ve*, *si*, and *c1*, and the margin between bases of setae *se* and *c1* is indistinct. Bases of setae *c1* are situated slightly posterior to the level of setal bases *se*. Propodonotal setae *ve* and *si* are short and subequal in length. The hysteronotal shield is absent. Bases of setae *d1* are situated closer to *d2* than to *e2*. Setae *c1* are about 1.5 times longer than *c2*. Setae *d2* are distinctly longer than *d1* and *e2*. The pygidial shield is apunctate and has a rounded anterior margin. The genital plate is absent. Genital setae *g1* and *g2* and pseudanal setae *ps1* are equal in length. Coxal fields I–II are well sclerotised, III–IV weakly sclerotised, and all apunctate. Body cuticular striations as in Figure 1. Legs. Solenidia of legs I as in Figure 2B. Fan-like setae of legs III and IV are with six tines (Figure 2C). Lengths of setae: *ve* 20 (20–30), *si* 20 (25–30), *se* 210 (185–210), *c1* 265 (270), *c2* 180 (190–225), *d1* 20 (15–25), *d2* 150 (145–195), *e2* 25 (20–30), *f1* (20–25), *f2* 50 (40–50), *h1* 20 (15–30), *h2* 415 (430–470), *ag1* 60 (50–70), *ag2* (30–45), *ag3* (90–100), *ps1* 20 (15–20), *g1* and *g2* 20 (15–20), *3b* 25 (20), *3c* 45 (40), *l’RI* 5 (10), *l’RII* (15), *l’RIII* (20–30), *l’RIV* 15 (20), *tc’III–IV* 40 (35–40), and *tc”III–IV* 65 (60–70).

Male. (Figure 2D–G). The total body length is 380–385 in five paratypes. Gnathosoma. The infracapitulum is apunctate. Movable cheliceral digits are 100–105 long. The stylophore is 145–150 long; the exposed portion of the stylophore has striae ornament, and is 115–120 long. Each medial branch of the peritremes has two chambers, and each lateral branch has eight or nine chambers (Figure 2D). Idiosoma. The propodonotal shield is trapezoidal in shape, weakly sclerotised and apunctate, bearing bases of setae *ve*, *si* and *c1*. Propodonotal setae *ve* and *si* are subequal in length. Bases of setae *se* are situated distinctly anterior to the level of setae *c1*. The hysteronotal and the pygidial shields are absent. Setae *d2* are 1.4–2 times longer than *d1* and *e2*. Setae *h2* are about twice as long as *f2*. Coxal fields are weakly sclerotised and apunctate. Body cuticular striations as in Figure 2G. The lengths of setae are as follows: *ve* 15–20, *si* 15–20, *se* 25–30, *c1* 30–35, *c2* 30–35, *d1* 15–20, *d2* 20–30, *e2* 15–20, *f2* 15–20, *h2* 30–35, *ag1* 30–40, *ag2* 20–30.

##### Type Material

Female holotype and paratypes: 45 females and five males (reg. no. AMU MS 21-0910-060) from the Common Hill Myna *Gracula religiosa* Linnaeus (host at SNSB-ZSM, uncatalogued); Indonesia, Malay Archipelago, Java, 1908, coll. W. Elbert.

##### Type Material Deposition

Holotype and most paratypes are deposited in the SNSB-ZSM, except 20 females and three males in the AMU.

##### Additional Material

Eleven females (reg. no. AMU MS 21-0910-052) from the White-necked Myna *Streptocitta albicollis* (Vieillot) (host at SNSB-ZSM, uncatalogued); Indonesia, Malay Archipelago, Celebes Isl., 1875, coll. Riedel.

##### Differential Diagnosis

*Aulonastus indonesianus* sp. n. is morphologically similar to the recently described *Aulonastus darwini* Skoracki, Sikora, Unsoeld and Hromada, 2022 collected from two host species of the genus *Geospiza* (Thraupidae) [35]. In females of both species, the infracapitulum is apunctate; setae *ve* and *si* are subequal in length; setae *c1* are longer than *se*; the genital plate is absent; fan-like setae have six or seven tines, and all coxal fields are apunctate. This new species differs from *A. darwini* by the following features: in females of *A. indonesianus*, the stylophore is 170–185 long; each lateral branch of the peritremes has seven chambers; the propodonotal shield bearing bases of setae *ve*, *si* and *c1*; bases of setae *c1* are situated slightly posterior to level of setal bases *se*, and the hysteronotal shield is absent. In females of *A. darwini*, the stylophore is 130–135 long; each lateral branch of the peritremes has four or five chambers; the propodonotal shield bearing bases of setae *ve*, *si*, *se* and *c1*; bases of setae *c1* and *se* are situated at the same transverse level, and the hysteronotal shield is present and fused with the pygidial shield.

##### Etymology

The name “*indonesianus*” is taken from the region where the hosts were captured—Indonesia.

#### 3.1.2. *Aulonastus anais* Skoracki and Patan sp. n.

Female, holotype (Figure 3 and Figure 4A–C). The total body length is 590 (550–600 in 11 paratypes). Gnathosoma. The infracapitulum is densely punctate. Movable cheliceral digits are 120 (120–125) long. The stylophore is 170 (165–170) long; the exposed portion of the stylophore is apunctate and 130 (125–130) long. Each medial branch of the peritremes has one or two chambers, and each lateral branch has four chambers (Figure 4A). Idiosoma. The propodonotal shield is well sclerotised, bearing bases of setae *ve*, *si* and *c1*, sparsely punctate near bases of setae *ve* and *si*, and the margin between bases of setae *se* and *c1* is indistinct. Bases of setae *c1* and *se* are situated at the same transverse level. Propodonotal setae *ve* and *si* are short and subequal in length, or setae *si* are slightly (1.3 times) longer than *ve*. Bases of setae *d1* are situated closer to *d2* than to *e2*. Setae *c1* are 1.5 times longer than *c2*. Setae *d2* are distinctly longer than *d1* and *e2*. The hysteronotal shield is narrow and apunctate, not fused with the pygidial shield, situated between bases of setae *d1* and *e2*. The pygidial shield is apunctate and has an indistinct anterior margin. The genital plate is absent. Genital setae *g1* and *g2* are equal in length. The coxal fields I–II are well sclerotised, III–IV weakly sclerotised, all punctate. Body cuticular striations as in Figure 3. Legs. Solenidia of legs I as in Figure 4B. Fan-like setae of legs III and IV are with eight tines (Figure 4C). Lengths of setae: *ve* 15 (15–20), *si* 20 (20–25), *se* 220 (205–235), *c1* 245 (220–240), *c2* 165 (150–175), *d2* 110 (110–130), *d1* 20 (20–25), *e2* 35 (35–50), *f1* 30 (30–35), *f2* 65 (50–65), *h1* 30 (30–35), *h2* (370–400), *ag1* 50 (50–60), *ag2* 35 (30–40), *ag3* 95 (80–110), *ps1* 25 (20–25), *g1* and *g2* 25 (25), *tc’III*–*IV* 35 (35), *tc”III*–*IV* 55 (55–60), *3b* 20 (20), *3c* 35 (30–35), *4c* 35 (30–35), *l’RIII* 25 (20–25), and *l’RIV* 20 (20).

Male. (Figure 4D,E). The total body length is 380 in one paratype. Gnathosoma. Infracapitulum is apunctate. The stylophore is 135 long; the exposed portion of the stylophore is with striae ornament and apunctate and is 110 long. Each medial branch of the peritremes has three chambers, and each lateral branch has five or six chambers (Figure 4D). Idiosoma. All dorsal shields are weakly sclerotised and apunctate. The propodonotal shield is trapezoidal in shape, bearing bases of setae *ve*, *si* and *c1*. Setae *ve* and *si* are subequal in length. Bases of setae *c1* and *se* are situated at the same transverse level. The hysteronotal shield is fused to the pygidial shield; the anterior margin is concave and reaches the level of setal bases *d1*. Setae *d1*, *d2* and *e2* are subequal in length. Setae *h2* are about three times longer than *f2*. Coxal fields are weakly sclerotised and apunctate. Body cuticular striations as in Figure 4E. Legs. The fan-like setae of legs III and IV have five or six tines. Lengths of setae: *ve* 10, *si* 10, *se* 25, *c1* 45, *c2* 25, *d1* 10, *d2* 15, *e2* 10, *f2* 15, and *h2* 50.

##### Type Material

Female holotype and paratypes: 11 females and one male (reg. no. AMU MS 21-0910-045) from the Golden Myna *Mino anais* (Lesson) (host reg. no. SNSB-ZSM 11.602; female); Papua New Guinea, August 1910, coll. L. von Wiedenfeld.

##### Type Material Deposition

The holotype and most paratypes are deposited in the SNSB-ZSM, except five females and one male in the AMU.

##### Differential Diagnosis

This new species is morphologically similar to the above described species, *A. indonesianus*, and can be easily distinguished by the following features: in females of *A. anais*, the infracapitulum is densely punctate; each lateral branch of the peritremes has four chambers; bases of setae *c1* and *se* are situated at same transverse level; the hysteronotal shield reduced to small and narrow shield, situated between bases of setae *d1* and *e2*; and all coxal fields are punctate. In females of *A. indonesianus*, the infracapitulum is apunctate; each lateral branch of the peritremes has seven chambers; bases of setae *c1* are situated slightly posterior to the level of setal bases *se*; the hysteronotal shield is absent; and all coxal fields are apunctate.

##### Etymology

The name “*anais*” is taken from the species name of the host, *Mino anais*.

#### 3.1.3. *Syringophiloidus poeopterus* Skoracki and Patan sp. n.

Female, holotype (Figure 5 and Figure 6A). The total body length is 675 (620–670 in three paratypes). Gnathosoma. The infracapitulum is apunctate. The stylophore has a length of 160 (155–160), and the length of the exposed portion of the apunctate stylophore is 130 (125–130). Each medial branch of the peritremes has two chambers, and each lateral branch has ten chambers (Figure 6A). Idiosoma. The propodonotal shield is well sclerotised, rectangular in shape, and sparsely punctate between bases of setae *ve* and *si*. Propodonotal setae *vi*, *ve*, and *si* are short, smooth, and subequal in length. Bases of setae *c1* and *se* are situated at the same transverse level. The hysteronotal shield is well sclerotised and apunctate; the anterior margin reaches above the level of setal bases *d2*, and the posterior margin is not fused to the pygidial shield and not reaching bases of setae *e2*. Bases of setae *d1* are situated closer to *d2* than to *e2*. Setae *d1*, *d2*, and *e2* are subequal in length. The pygidial shield is apunctate and with rounded anterior margin. The genital plate is absent. Agenital setae *ag1*–*3* are subequal in length. Genital setae *g1* and *g2* are equal in length. Pseudanal setae *ps1* and *ps2* are equal in length. Coxal fields I–IV are well sclerotised, I–II are sparsely punctate or apunctate, and III–IV are punctate. They have body cuticular striations, as shown in Figure 5. Legs. The fan-like setae of legs III and IV have eight or nine tines. The lengths of setae are as follows: *vi* 30 (25–30), *ve* 30 (30–35), *si* 30 (30–40), *se* 195 (190–200), *c1* 185 (180–195), *c2* 220 (195–215), *d1* 125 (115–130), *d2* (125–145), *e2* (130–135), *f1* 20 (20–25), *h1* 20 (20–25), *h2* 280 (270–305), *ag1* 120 (120–145), *ag2* 130 (110–135), *ag3* 150 (140–150), *ps1* and *ps2* 30 (25–30), *g1* and *g2* 30 (25–30), *4b* 25 (25–30), *4c* (80), *l’RIII* 35 (35–40), and *l’RIV* 25 (25).

Male (Figure 6B–E). The total body length is 475–560 in two paratypes. Gnathosoma. The infracapitulum is apunctate. The stylophore has a length of 160 (155–160), and the exposed portion of the stylophore is apunctate and 135–145 long. Each medial branch of the peritremes has two or three chambers, and each lateral branch has ten chambers (Figure 6B,C). Idiosoma. The propodonotal shield is well sclerotised, rectangular in shape, and sparsely punctate between bases of setae *ve* and *si*. Propodonotal setae *vi*, *ve*, and *si* are short, smooth, and subequal in length. Bases of setae *c1* are situated posterior to the level of setae *se*. The hysteronotal shield is well sclerotised, large, and punctate, the anterior margin reaching above the level of setal bases *d2*, the posterior margin is not fused to the pygidial shield and reaching bases of setae *e2*. Bases of setae *d1* are situated closer to *d2* than to *e2*. Setae *d2* are 2.5 times longer than *d1* and *e2*. The pygidial shield has indistinct anterior margin. Agenital setae *ag1* are 1.4 times longer than *ag2*. The body cuticular striations as in Figure 6E. The lengths of setae are as follows: *vi* 20–35, *ve* 25–40, *si* 30–35, *se* 130–145, *c1* 115–140, *c2* 100–150, *d1* 10, *d2* 20–25, *e2* 10, *f2* 10–20, *h2* 110–120, *ag1* 50–75, and *ag2* 35–40.

##### Type Material

Female holotype, three female paratypes and two male paratypes (reg. no AMU MS 21-1012-042) from the quill of wing covert of the Abbott’s Starling *Poeoptera femoralis*; Tanzania, Arusha Region, Arusha National Park, Mt. Meru, 2164 m a.s.l., 16 November 1958, coll. Nagy.

##### Types Deposition

Holotype deposited in the SNSB-ZSM, paratypes in the AMU.

##### Differential Diagnosis

This new species is morphologically similar to *Syringophiloidus saponai* Skoracki, Patan and Unsoeld, 2024 recorded from four host species of the genus *Lamprotornis* in Kenya, Tanzania and Ethiopia [36], by the presence of short setae *vi*, *ve*, and *si* (all shorter than 70). *S. poeopterus* can be easily distinguished from *S. saponai* by the shorter hysteronotal setae *d1*, *d2*, and *e2*, which are 115–130, 125–145, and 130–135, respectively (*vs* the length of setae *d1*, *d2*, and *e2* are 190–205, 260–285, and 200–215, respectively, in *S. saponai*). Additionally, in females of *S. poeopterus*, a genital plate is absent (*vs* genital plate is present in *S. saponai*).

##### Etymology

The name “*poeopterus*” is taken from the generic name of the host, *Poeoptera*.

#### 3.1.4. *Syringophiloidus presentalis* Chirov and Kravtsova, 1995

Host and distribution. Sturnidae: the Eurasian Starling *Sturnus vulgaris* Linnaeus from Kyrgyzstan [37], Poland, Slovakia, and France [4].

#### 3.1.5. *Syringophiloidus saponai* Skoracki, Patan and Unsoeld, 2024

Hosts and distribution. Sturnidae: the Greater Blue-eared Glossy-Starling *Lamprotornis chalybaeus* Hemprich and Ehrenberg from Kenya, Tanzania, and Ethiopia; the Superb Starling *Lamprotornis superbus* Rüppell from Tanzania and Kenya; the Lesser Blue-eared Glossy-Starling *Lamprotornis chloropterus* Swainson from Tanzania; the Ashy Starling *Lamprotornis unicolor* (Shelley) from Tanzania [36]; the Pale-winged Starling *Onychognathus nabouroup* (Daudin) from Namibia, and the Red-winged Starling *Onychognathus morio* (Linnaeus) from Tanzania (current study).

##### New Material Examined

Four females and one male (reg. no AMU MS 21-1012-031) from the Pale-winged Starling *Onychognathus nabouroup* (Daudin) (host reg. no. SNSB-ZSM 57.20); Namibia, 5 November 1956, no other data. Five females and one male (reg. no. AMU MS 21-1012-034) from the Red-winged Starling *Onychognathus morio* (Linnaeus) (host reg. no. SNSB-ZSM 64.717); Tanzania, Morogoro District, 6 May 1962, coll. Th. Andersen.

#### 3.1.6. *Syringophiloidus graculae* Fain, Bochkov and Mironov, 2000

Hosts and distribution. Sturnidae: the Common Hill Myna *Gracula religiosa* Linnaeus from SE Asia [38]; the Golden-crested Myna *Ampeliceps coronatus* Blyth from Indochina, and the Coleto *Sarcops calvus* (Linnaeus) from the Philippines (current study).

##### New Material Examined

Seventeen females and two males (reg. no. ZISP AVB 05-0726-010) from the Golden-crested Myna *Ampeliceps coronatus* Blyth; Indochina, no other data. Twelve females and one male (reg. no. AMU MS 21-0910-054) from the Coleto *Sarcops calvus* (Linnaeus) (host reg. no. SNSB-ZSM 26.213); Philippines, Cebu Isl., 1879, coll. Burger. Four females (reg. no. AMU MS 21-0910-055) from the same host species (host reg no. SNSB-ZSM 26.215) and locality.

#### 3.1.7. *Syringophilopsis sturni* Chirov and Kravtsova, 1995

Host and distribution. Sturnidae: the Eurasian Starling *Sturnus vulgaris* Linnaeus from Kyrgyzstan [37], Kazakhstan [39], Poland [40], and Ukraine [4].

#### 3.1.8. *Syringophilopsis parasturni* Skoracki, Patan and Unsoeld, 2024

Hosts and distribution. Sturnidae: the Chestnut-bellied Starling *Lamprotornis pulcher* (Müller) from Senegal; the Bronze-tailed Glossy-Starling *Lamprotornis chalcurus* Nordmann from Senegal [36]; the Black-bellied Glossy-Starling *Notopholia corusca* (Nordmann) from Tanzania, and the Abbott’s Starling *Poeoptera femoralis* (Richmond) from Tanzania (current study).

##### New Material Examined

Seven females and two males (reg. no. AMU MS 22-0821-012) from the Black-bellied Glossy-Starling *Notopholia corusca* (Nordmann) (org. *Lamprotornis corruscus*) (host in the SNSB-ZSM, uncatalogued); Tanzania, Tanga Region, Tanga, March 1893, coll. O. Neumann. Seven females and one male (reg. no. AMU MS 21-1012-041) from the Abbott’s Starling *Poeoptera femoralis* (Richmond) (host in the SNSB-ZSM, uncatalogued); Tanzania, Arusha Region, Arusha National Park, Mt. Meru, 2560 m a.s.l., 2 November 1959, coll. Nagy. Nine females (reg. no AMU MS 21-1012-042) from the same host species (host reg, no SNSB-ZSM 59.148) and locality, 2164 m a.s.l., coll. Nagy.

#### 3.1.9. *Krantziaulonastus buczekae* (Skoracki, 2002)

Host and distribution. Sturnidae: the Eurasian Starling *Sturnus vulgaris* Linnaeus from Poland [41].

The world fauna of quill mites of the subfamily Syringophilinae associated with Starlings is summarised in Table 1.

## 4. Discussion

The bird family Sturnidae (Starlings and Mynas) includes approximately 125 species divided into 36 genera [42]. Their distribution is confined to the Old World, naturally occurring in Europe, Asia, Africa, northern Australia, and the Pacific islands, except for anthropogenic introductions and/or invasions in regions such as New Zealand and both Americas. The centres of biodiversity of this family are identified in Southeast Asia and Africa [42,43]. Our study has identified four genera of quill mites belonging to the subfamily Syringophilinae, which are prevalent across a wide array of passerine birds.

Mite species belonging to the genus *Syringophiloidus* have been recorded on hosts across 22 passerine families [4,44,45]. Currently, we have recorded four species residing on starlings observed across the Oriental, Palearctic, and Ethiopian regions, as well as on both basal and crown starling lineages [43]. This pattern suggests that the genus *Syringophiloidus* established a symbiotic relationship with starlings prior to their worldwide diversification, likely during the Miocene period [46,47].

Similarly, species from the genus *Syringophilopsis* predominantly inhabit passerine birds, having been documented in up to 27 families [44]. Although two species from this genus have only been recorded on starlings occurring in the Ethiopian and Palaearctic regions, the absence of records from the Oriental region may be due to insufficient specimen examination. It is conceivable that future research will uncover the presence of this genus in the Oriental region as well. Similar to *Syringophiloidus*, it is plausible that *Syringophilopsis* established its association with starlings before their global dispersal.

The distribution of mites from the genera *Aulonastus* (two species) and *Krantziaulonastus* (one species) presents a notable scenario. Representatives of the genus *Aulonastus* are found in the body feathers of the rather basal Asian jungle starlings’ lineage in the Oriental region [43]. In contrast, in Europe, the Eurasian Starling (*Sturnus vulgaris*), a member of the Eurasian savannah starlings’ clade, hosts a member of a different genus, *Krantziaulonastus*, occupying the same ecological niche. Interestingly, neither of these syringophiline genera has been recorded in Africa, where different crown starling clades have diversified. However, syringophilids belonging to the subfamily Picobiinae occupy their body feather quills, utilising the same niche.

## 5. Conclusions

Our study identified the presence of nine mite species belonging to four genera of the subfamily Syringophilinae. Our dataset elucidates the host specificity of these quill mite species, revealing a combination of mono- and oligoxenous tendencies. The latter are consistently restricted to hosts from specific zoogeographical regions and infest closely related genera. This specificity indicates a nuanced symbiotic relationship between quill mites and their avian hosts, likely shaped by intricate coevolutionary dynamics. In conclusion, our research not only enhances the knowledge of quill mite diversity and host specificity but also underscores the importance of continued studies to gain a more comprehensive understanding of the ecological and evolutionary aspects of these symbioses. Through this research, we gain a better understanding of how quill mites adapt to their hosts and how these interactions can influence biodiversity and ecosystem functioning.

## Figures and Tables

**Figure 1 animals-14-02239-f001:**
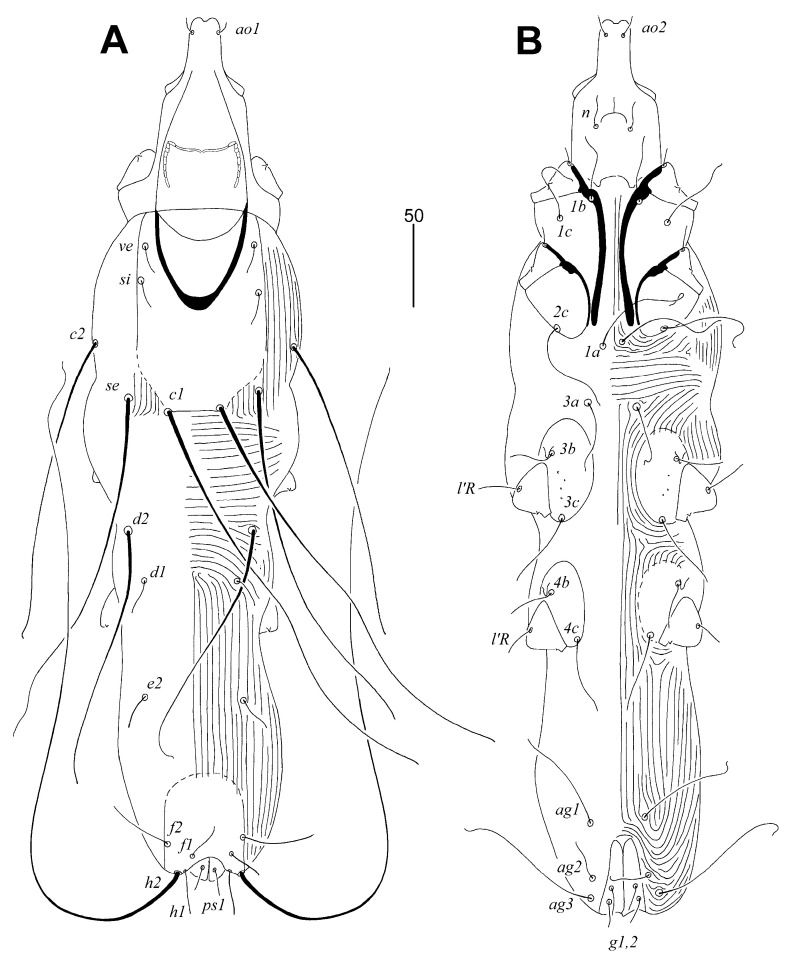
*Aulonastus indonesianus* Marcisova, Skoracki and Patan sp. n., female. (**A**)—dorsal view; (**B**)—ventral view.

**Figure 2 animals-14-02239-f002:**
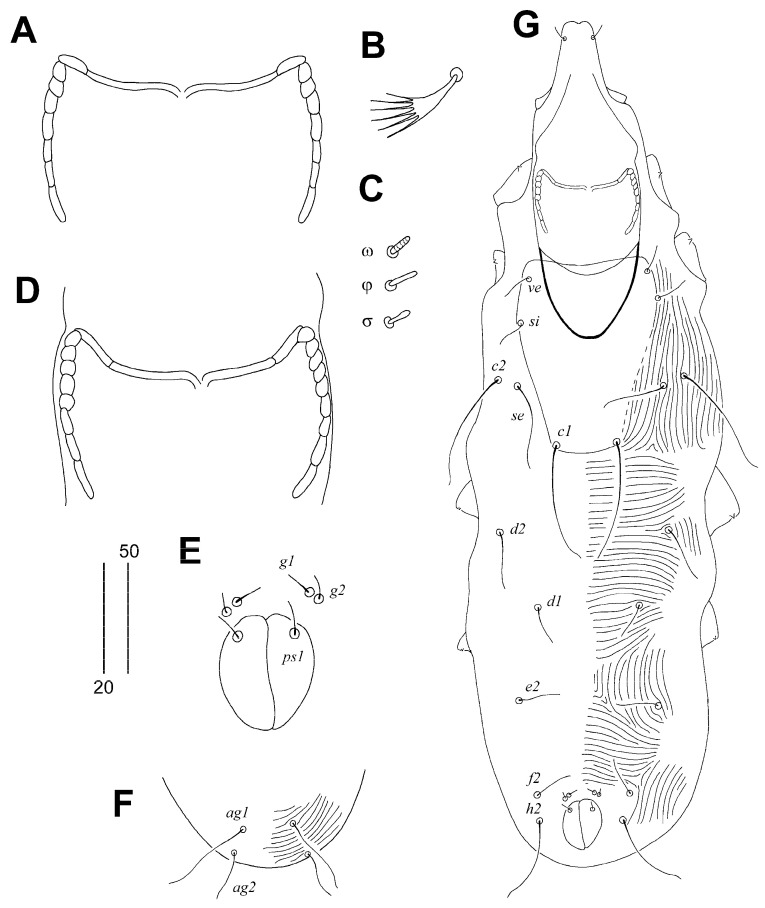
*Aulonastus indonesianus* Marcisova, Skoracki and Patan sp. n., female (**A**–**C**), male (**D**,**E**). (**A**)—peritremes; (**B**)—fan-like seta *p’III*; (**C**)—solenidia of leg I; (**D**)—peritremes; (**E**)—genito-anal opening; (**F**)—opisthosoma in ventral view; (**G**)—body in dorsal view. Scale bars: (**A**–**E**) = 20 µm, (**F**,**G**) = 50 µm.

**Figure 3 animals-14-02239-f003:**
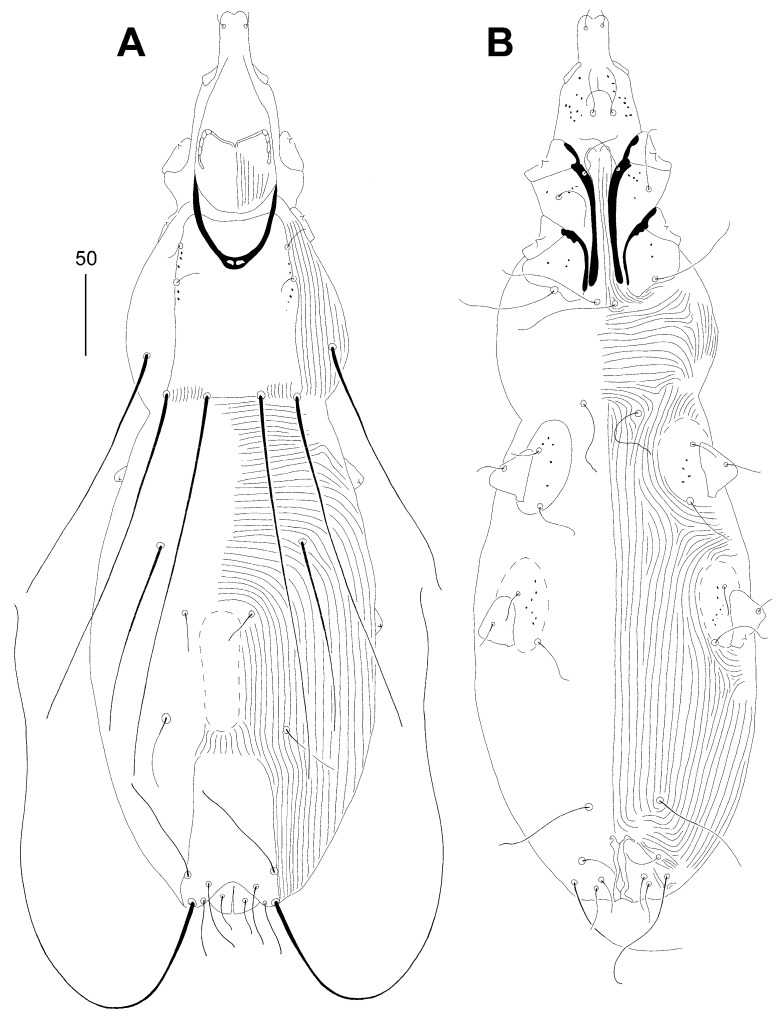
*Aulonastus anais* Skoracki and Patan sp. n., female. (**A**)—dorsal view; (**B**)—ventral view.

**Figure 4 animals-14-02239-f004:**
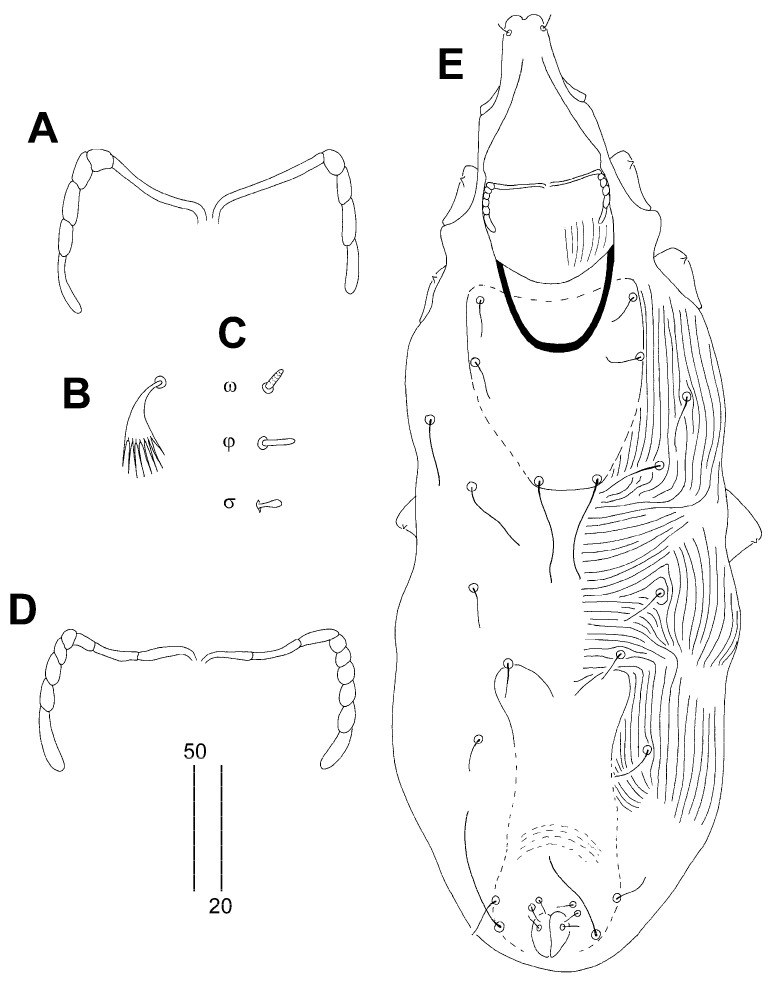
*Aulonastus anais* Skoracki and Patan sp. n., female (**A**–**C**), male (**D**,**E**). (**A**)—peritremes; (**B**)—fan-like seta p’III; (**C**)—solenidia of leg I; (**D**)—peritremes; (**E**)—body in dorsal view. Scale bars: (**A**–**D**) = 20 µm, (**E**) = 50 µm.

**Figure 5 animals-14-02239-f005:**
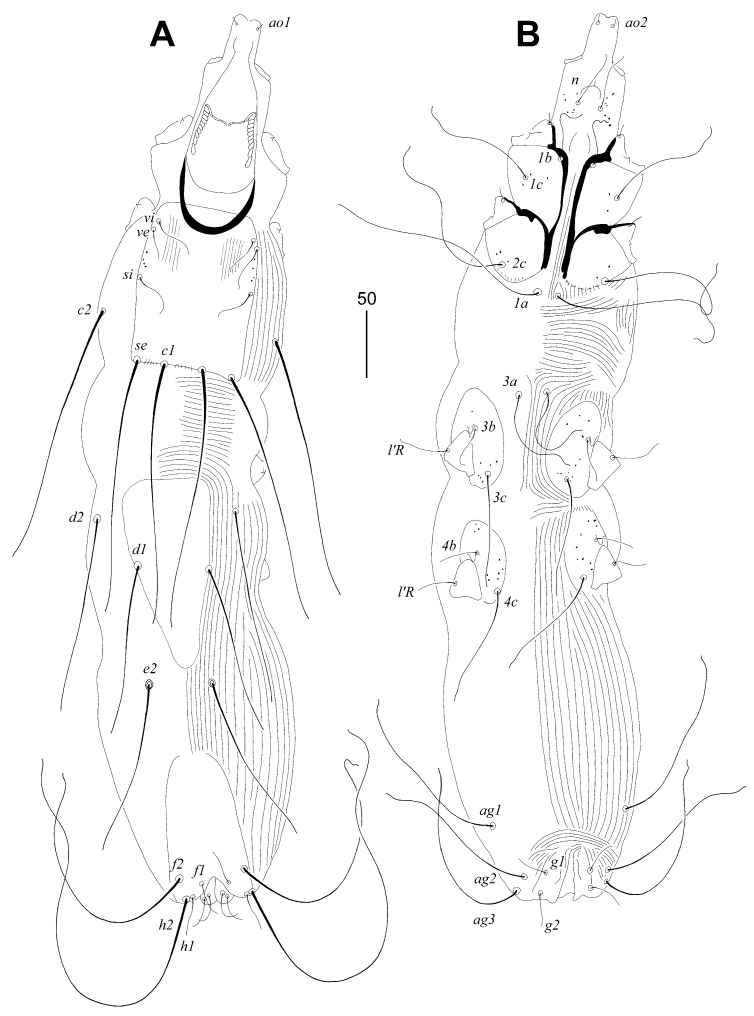
*Syringophiloidus poeopterus* Skoracki and Patan sp. n., female. (**A**)—dorsal view; (**B**)—ventral view.

**Figure 6 animals-14-02239-f006:**
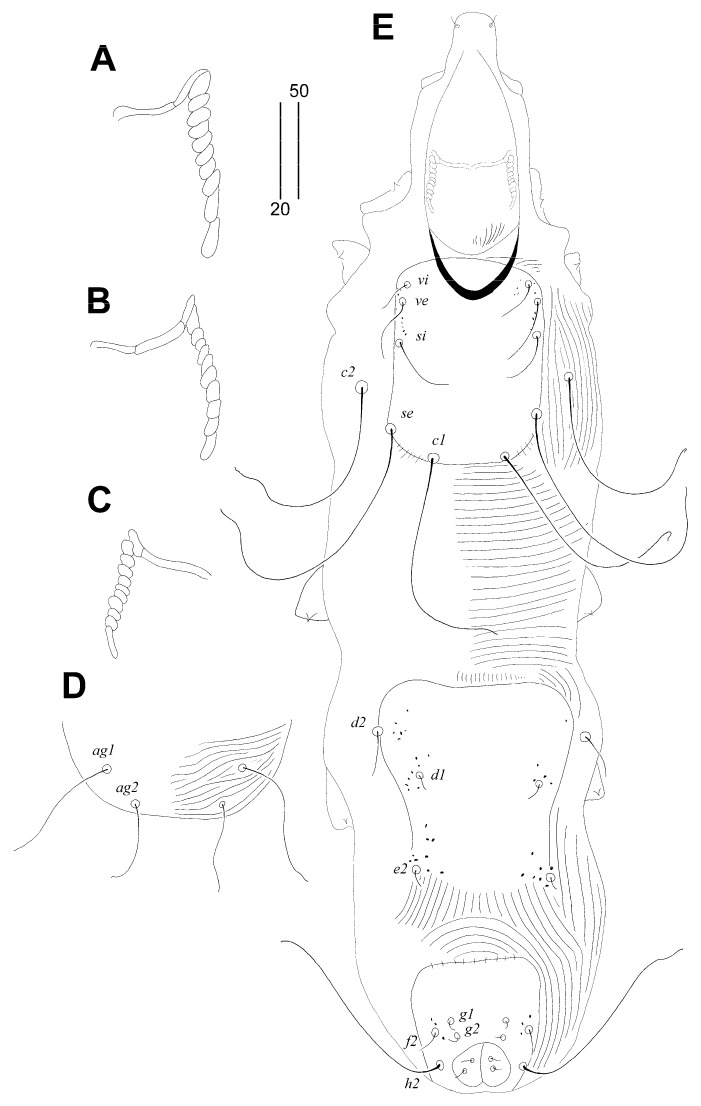
*Syringophiloidus poeopterus* Skoracki and Patan sp. n., female (**A**), male (**B**–**E**). (**A**)—peritremes; (**B**,**C**)—various types of peritremes; (**D**)—opisthosoma in ventral view; (**E**)—body in dorsal view. Scale bars: (**A**–**C**) = 20 µm, (**D**,**E**) = 50 µm.

**Table 1 animals-14-02239-t001:** Quill mite species of the subfamily Syringophilinae (fam. Syringophilidae) associated with starlings. Abbreviations: OR—Oriental, PM—Papua-Melanesian, ET—Ethiopian, PA—Palaearctic.

Mite Species	Host Species	Locality	References
*Aulonastus indonesianus* Marcisova, Skoracki and Patan sp. n.	Common Hill Myna, *Gracula* *religiosa* Linnaeus	OR: Indonesia (Java)	Current study
“	White-necked Myna, *Streptocitta albicollis* (Vieillot)	OR: Indonesia (Celebes)	Current study
*Aulonastus anais* Skoracki and Patan sp. n.	Golden Myna, *Mino anais* (Lesson)	PM: Papua New Guinea	Current study
*Krantziaulonastus buczekae* Skoracki, 2002	Common Starling, *Sturnus* *vulgaris* Linnaeus	PA: Poland	[4,41]
*Syringophiloidus poeopterus* Skoracki and Patan sp. n.	Abbott’s Starling, *Poeoptera femoralis* (Richmond)	ET: Tanzania	Current study
*Syringophiloidus presentalis* Chirov and Kravtsova, 1995	Common Starling, *Sturnus* *vulgaris* Linnaeus	PA: Poland, Slovakia, France, Kyrgyzstan	[4,37]
*Syringophiloidus saponai* Skoracki, Patan and Unsoeld, 2024	Greater Blue-eared Glossy-Starling, *Lamprotornis chalybaeus* Hemprich and Ehrenberg	ET: Kenya, Tanzania, Ethiopia	[36]
“	Superb Starling, *Lamprotornis* *superbus* Rüppell	ET: Tanzania, Kenya	[36]
“	Lesser Blue-eared Glossy-Starling, *Lamprotornis chloropterus* Swainson	ET: Tanzania	[36]
“	Ashy Starling, *Lamprotornis* *unicolor* (Shelley)	ET: Tanzania	[36]
“	Pale-winged Starling, *Onychognathus nabouroup* (Daudin)	ET: Namibia	Current study
“	Red-winged Starling, *Onychognathus morio* (Linnaeus)	ET: Tanzania	Current study
*Syringophiloidus graculae* Fain, Bochkov and Mironov, 2000	Common Hill Myna, *Gracula* *religiosa* Linnaeus	OR: SE Asia	[38]
“	Golden-crested Myna, *Ampeliceps coronatus* Blyth	OR: Indochina	Current study
“	Coleto, *Sarcops calvus* (Linnaeus)	OR: Philippines	Current study
*Syringophilopsis sturni* Chirov and Kravtsova, 1995	Common Starling, *Sturnus* *vulgaris* Linnaeus	PA: Kyrgyzstan, Kazakhstan, Poland, Ukraine	[4,37,39,40]
*Syringophilopsis parasturni* Skoracki, Patan and Unsoeld, 2024	Chestnut-bellied Starling, *Lamprotornis pulcher* (Müller)	ET: Senegal	[36]
“	Bronze-tailed Glossy-Starling, *Lamprotornis chalcurus* Nordmann	ET: Senegal	[36]
“	Black-bellied Glossy-Starling, *Notopholia corusca* (Nordmann)	ET: Tanzania	current study
“	Abbott’s Starling, *Poeoptera femoralis* (Richmond)	ET: Tanzania	current study

## Data Availability

All necessary data (such as localities) are available in the text of this article.

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
