# Peer review of "Quill Mites of the Subfamily Syringophilinae (Acariformes: Syringophilidae) Parasitising Starlings (Passeriformes: Sturnidae)†"

_animals, 2024, doi:10.3390/ani14152239_

Round 1

Reviewer 1 Report

Comments and Suggestions for Authors

Dear authors

A major revision is advised, as follows

- L2-L3 Please complete the title adding information of the origin of the studied birds (i.e ".... : Sturnidae from Tanzania, Namibia, Indonesia, Philippines and Papua New Guinea in the museum of the Bavarian State Collection of Zoology, Munich, Germany")

- L20 Please clarify what starling hosts you are referring to.

- L55-L56 Please complete the sentence including body areas most commonly affected by pecking. What aspect you see in these parasitized feathers?  

- L61 Please expand the bacterial species potentially carried by these mites.

- L63-L65 The authors highlighted the vertical transmission of mites without supporting the sentence with a cited reference. Are you referring to natural or experimentally induced infections? Please add a reference of it. A description of horizontal transmission is needed also.  

- L89 to L93 Please revise and rewrite the aim of this work in order to be concise and informative.  

- L96 to L101 Please add information of the studied starlings included in this work such as common and scientific names, age, sex, year of capture, presence of absence of lesions in skin and feathers, etc.

- L99 to L101 Do you refer to all carcass of the studied birds? Can you briefly describe the protocol named here?

- L103 Please add information about the light microscope used in this study (Trademark, Country and city of origin, etc.).

- L104-L107 Please rewrite this paragraph to clarify this methodology.

- L108 to L112 Please rewrite to clarify 

- Nanometer (nm) should be included in every single measurement made, to avoid any misunderstanding during the reading

- Table 1

. L381 The term "associated with" is not correct. Please use "parasitizing" instead.

. Please include common names of the starlings.

- L395 Please change from "[see 55]' to "[55]". 

- L395 to L397 You are referring to a symbiotic relationship between starlings and quill mites before worldwide diversification. Could you be more specific about this timeline? This point is one relevant features of your manuscript, and information in the Introduction is advised.

- The Conclusion section can be improved in length.

Comments on the Quality of English Language

- L86 Please use past tense to refer to previously published findings.

Author Response

We would like to thank the Reviewer for taking the time to read our manuscript and for providing all the comments and suggestions. Our responses to the Reviewer's suggestions are presented below:

Comments: L2-L3 Please complete the title adding information of the origin of the studied birds (i.e ".... : Sturnidae from Tanzania, Namibia, Indonesia, Philippines and Papua New Guinea in the museum of the Bavarian State Collection of Zoology, Munich, Germany")

Response: Thank you very much for the suggestion. However, we believe that the title we proposed is appropriate. The manuscript presents a comprehensive revision of mites from the subfamily Syringophilinae that parasitize birds of the family Sturnidae. It includes the description of new species as well as new records for previously described species.

Comments: L20 Please clarify what starling hosts you are referring to.

Response: It is presented in the abstract. We omitted this data in the simple summary.

Comments: L55-L56 Please complete the sentence including body areas most commonly affected by pecking. What aspect you see in these parasitized feathers?

Response: This is a suggestion from the author Gritschenko, not from us.

Comments: L61 Please expand the bacterial species potentially carried by these mites.

Response: Thank you for this suggestion. We have added this information.

Comments: L63-L65 The authors highlighted the vertical transmission of mites without supporting the sentence with a cited reference. Are you referring to natural or experimentally induced infections? Please add a reference of it. A description of horizontal transmission is needed also.

Response: These details are included in the first part of the text, lines 41-45, along with the citations.

Comments: L89 to L93 Please revise and rewrite the aim of this work in order to be concise and informative.

Response: Thank you. We have rewritten this part of the text.

Comments: L96 to L101 Please add information of the studied starlings included in this work such as common and scientific names, age, sex, year of capture, presence of absence of lesions in skin and feathers, etc.

Response: Thank you, we added this information to the text.

Comments: L99 to L101 Do you refer to all carcass of the studied birds? Can you briefly describe the protocol named here?

Response: Thank you. We added this information to the text.

Comments: L103 Please add information about the light microscope used in this study (Trademark, Country and city of origin, etc.).

Response: Thank you, we added this information to the text.

Comments:  L104-L107 Please rewrite this paragraph to clarify this methodology

Response: Thank you. We have rewritten this part of the text.

Comments:  L108 to L112 Please rewrite to clarify

Response: Thank you, we have rewritten this part of the text.

Comments:  Nanometer (nm) should be included in every single measurement made, to avoid any misunderstanding during the reading.

Response: It is unnecessary to include this in every measurement presented in the descriptions. In the Materials and Methods section, we clearly stated that all measurements presented in the species descriptions are given in micrometres.

Comments: L381 The term "associated with" is not correct. Please use "parasitizing" instead.

Response: Thank you, we corrected it.

Comments: Please include common names of the starlings.

Response: We added the common names for starlings in table 1.

Comments: L395 Please change from "[see 55]' to "[55]". 

Response: Thank you, we corrected it.

Comments: L395 to L397 You are referring to a symbiotic relationship between starlings and quill mites before worldwide diversification. Could you be more specific about this timeline? This point is one relevant features of your manuscript, and information in the Introduction is advised.

Response: Thank you, we added this information to the text.

Comments: The Conclusion section can be improved in length.

Response: Thank you, we improved the chapter Conclusion.

Comments:  L86 Please use past tense to refer to previously published findings.

Response: Thank you, we corrected it.

Reviewer 2 Report

Comments and Suggestions for Authors

Dear Authors,

This is a well-produced paper. The English are very good, the literature used is adequate and it is an interesting paper, I made small corrections in the paper and will send that corrected manuscript this to the editor. As the contents of this paper are common knowledge to you I have only two suggestions, mainly for beginners who may consult this paper. In the text, I suggested you indicate the positions of the chelicerae and the stylophore in the figures. Congratulations.

Author Response

Dear Reviewer,

Thank you very much for taking the time to read our manuscript and for all the corrections made directly on it. We have considered all of them. Regarding the chelicerae and stylophore, please see the general morphology of syringophilids described by Kethley (1970), or Skoracki (2011), or Skoracki et al. (2016). We would be glad to send these publications if you wish.

Reviewer 3 Report

Comments and Suggestions for Authors

Dear editor

Manuscript ID

Animals-3112096

The manuscript "Quill mites of the subfamily Syringophilinae (Acariformes: Syringophilidae) parasitizing starlings (Passeriformes: Sturnidae)" has been evaluated and needs some modifications. Overall, the manuscript is well written and presented, and describes three new species of mites. In the introduction, I suggest authors rearrange some paragraphs. For example, lines 66-81 I suggest combining into a single paragraph, more objective writing. The writing of the Materials and Methods is poor and requires more details. For example, host age, sex, collection time, capture coordinates, are important data. How the mites were collected, whether they were intact or whether it was necessary to hydrate them for the assembly of the blades.

Introduction

Line 39: Delete "of the family"

line 68: "on birds"

 Materials and Methods

Line 96: Delete "stuff"

Line 98: "Before mounting" the sentence is not clear.

Lines: 108-110: "Specimen depositories are cited using the following abbreviations: AMU – Adam Mickiewicz University, Department of Animal Morphology, PoznaÅ„, Poland; SNSB-ZSM – Bavarian State Collection of Zoology, Munich, Germany" insert deposit numbers .

Table 1: enter coordinates, sex and age of the birds.

Conclusions

I suggest rewriting, mention the three new species too.

Author Response

Dear Reviewer,

Thank you very much for taking the time to read our manuscript and for providing all the comments and suggestions. Our responses to the suggestions are presented below:

Comments: lines 66-81 I suggest combining into a single paragraph, more objective writing.

Response: Thank you. We have rewritten this part of the text.

Comments: Materials and Methods is poor and requires more details. For example, host age, sex, collection time, capture coordinates, are important data. How the mites were collected, whether they were intact or whether it was necessary to hydrate them for the assembly of the blades.

Response: We improved the chapter "Materials and methods".

Comments: line 68: "on birds"

Response: We corrected it.

Comments: Line 98: "Before mounting" the sentence is not clear.

Response: We corrected it.

Comments: Lines: 108-110: "Specimen depositories are cited using the following abbreviations: AMU – Adam Mickiewicz University, Department of Animal Morphology, PoznaÅ„, Poland; SNSB-ZSM – Bavarian State Collection of Zoology, Munich, Germany" insert deposit numbers .

Response: All registration numbers for the collected material are presented in the "Type material" and "Additional material" sections in each species description.

Comments: Conclusions, I suggest rewriting, mention the three new species too.

Response: Thank you, we improved the conclusion.

Reviewer 4 Report

Comments and Suggestions for Authors

The manuscript contains interesting data on mites from the Syringophilidae family, widely distributed parasites of birds. The work provides valuable information, including data on the host range and distribution of six known species of these mites associated with starlings (with new records of hosts and locations). Particularly valuable are descriptions of three species new to science from Indonesia, Papua New Guinea and Tanzania. Parasitological data, including parasite-host interactions, were also analyzed.

The research was carried out methodologically correctly, and its scientific reliability is beyond any doubt. The only note - in the "Material and methods", the information should be provided in detail and precisely, in accordance with the editorial guidelines. Meanwhile, mainly the methodology was described, but does not indicate how much material was examined (some of this data is scattered in the Results). It is worth specifying how many specimens/species of birds (how large a sample) were included in the research.

The work contains a relatively large number of self-citations, but they are justified and result from the leading role of the authors in global research on this group of mites.

Author Response

Dear Reviewer,

Thank you very much for taking the time to read our manuscript and for providing all the comments and suggestions. Our responses to the suggestions are presented below:

Comments: In the "Material and methods", the information should be provided in detail and precisely, in accordance with the editorial guidelines. Meanwhile, mainly the methodology was described, but does not indicate how much material was examined (some of this data is scattered in the Results). It is worth specifying how many specimens/species of birds (how large a sample) were included in the research.

Response: Thank you for the suggestion. We improved the chapter "Material and methods".

Comments: The work contains a relatively large number of self-citations, but they are justified and result from the leading role of the authors in global research on this group of mites.

Response: We fully agree with the reviewer's opinion.

Round 2

Reviewer 1 Report

Comments and Suggestions for Authors

Dear authors

Thank for addressing most of edits, changes, and suggestions previously made. I have no further comments to make.